# The cross-sectional association between mean corpuscular volume level and cognitive function in Chinese over 45 years old: Evidence from the China Health and Retirement Longitudinal Study

Yao Chen[1☯], Chen'Xi' Nan Ma[2☯], Lan Luo[2], Jieyun Yin[2], Zhan Gao[3], Zengli Yu[1]*, Zhongxiao Wan [ID][1,2]*

1 College of Public Health, Zhengzhou University, Zhengzhou, Henan, China, 2 School of Public Health, Medical College of Soochow University, Suzhou, China, 3 The Fifth Affiliated Hospital, Zhengzhou University, Zhengzhou, Henan, China

☯ These authors contributed equally to this work.
* zhxwan@suda.edu.cn (ZW); zly@zzu.edu.cn (ZY)

**Data Availability Statement:** All relevant data are within the manuscript.

## Abstract

Fewer studies have focused on the independent association between mean corpuscular volume (MCV) and cognitive performance. This study was designed to characterize the cross-sectional association between MCV and cognitive performance in a large sample of Chinese residents (age≥45 years) from the China Health and Retirement Longitudinal Study (CHARLS). A total of 4023 male and 4173 female adults with MCV ≥ 80 fl were included for analysis. By multivariable linear regression analysis, for the total subjects, MCV level was significantly negatively associated with global cognitive function and episodic memory. When adjusted by sex, only in male subjects, higher MCV level was associated with reduced scores for global cognitive function, episodic memory and mental status. Via binary logistic regression analysis, the higher MCV level (MCV>100 fl) was associated with poor global cognitive function (OR = 1.601; 95% CI = 1.198–2.139; p = 0.001), episodic memory (OR = 1.679; 95% CI = 1.281–2.201; p<0.001), and mental status (OR = 1.422; 95% CI = 1.032–1.959; p = 0.031) for the whole participants. When testing this association by sex, the significant relationship between higher MCV level with worse episodic memory was observed both in male (OR = 1.690; 95% CI = 1.211–2.358; p = 0.002) and female (OR = 1.729; 95% CI = 1.079–2.770; p = 0.023) subjects; while the association between higher MCV level and poor global cognitive function (OR = 1.885; 95% CI = 1.329, 2.675; p<0.001) and mental status (OR = 1.544; 95% CI = 1.034, 2.306; p = 0.034) only existed in male subjects. Further studies are warranted to clarify the association between MCV level and cognitive performance by considering sex into consideration both cross-sectionally and longitudinally.

**Funding:** This study was supported by the National Natural Science Foundation of China (grant NO. 81872609).The funders had no role in study design, data collection and analysis, decision to publish, or preparation of the manuscript.

**Competing interests:** The authors have declared that no competing interests exist.

## Introduction

Dementia is one of the non-communicable diseases, which has exerted the greatest economic and social burden worldwide [1]. It is estimated by the WHO that 4.6 million new patients are inflicted with dementia each year with numbers affected nearly doubling every 20 years to reach 81.1 million by 2040 [2]. In the past 30 years, China's economy has developed exponentially and Chinese population has been aging rapidly [3]. A meta-analysis published in 2018 reported that the overall prevalence of dementia in Chinese elderly (>60 years old) was 5.30% (95%CI: 4.30, 6.30) [4]. Especially, as for one of the major form of dementia, i.e., Alzheimer's disease (AD), China bears a heavy burden of AD costs, which greatly change the evaluation of AD cost around the world [5]. Nevertheless, there are no effective approaches to slow down the rapid rate of dementia incidence. Dementia is the final stage of many years' development of pathological changes in the cerebral tissue [6]. In order to prevent dementia, it is of great significance to pay more attention to potential risk factors associated with cognitive performance, especially in Chinese population.

Multiple modifiable risk factors including anemia might be associated with cognitive decline or dementia. For example, results from previous studies have suggested that anemia and low hemoglobin concentrations are independent risk factors of cognitive decline [7–13]. Mechanistically, anemia could lead to brain hypo-oxygenation and consequently to cognitive decline [14]. In contrast, accumulating evidence also suggest that there was no association between anemia or low hemoglobin and cognitive performance or impairment after adjusting for a series of factors in different study populations [15–18]. Except for anemia and hemoglobin concentration, mean corpuscular volume (MCV) can reflect the morphology of erythrocyte, which is commonly used as an index of anemia [19]. Anemia is categorized into microcytic anemia (under 80 fl), normocytic anemia (80–100 fl) and macrocytic anemia(over 100 fl) based on the value of MCV [19]. Healthy erythrocyte ensures that oxygen- and nutrient-rich blood is pumped to the tissues especially brain tissue, so they can function normally. However, few studies have explored the independent association between MCV and cognitive function. The earlier study by Danon et al. [20] observed that a significantly negative correlation existed between erythrocyte volume and memory performance [20]. In recent years, Gamaldo et al. [21] found that high MCV level in older adults is associated with poorer cognitive function and this relationship appears not to be explained by anemia and inflammation by analyzing data from the Baltimore Longitudinal Study of Aging (BLSA). In contrast, by using data from AddNeuroMed Study, no significant linear relationship was observed between MCV and MMSE score [12]. Additionally, there is also evidence suggesting that biological sex plays a critical role in cognitive function [22] and that the relationship between abnormal hemoglobin with worse global cognition was greater in women than in men [13]. Therefore, it is of great necessity to further clarify the association between abnormal MCV level and cognitive performance by taking sex into consideration, also in Chinese subjects.

Owing to the fact that larger red blood cells may have more difficulty passing through small capillaries, compromising to deliver adequate amounts of oxygen to cerebral tissues, there is biological plausibility that higher MCV level than the normal level might be associated with worse cognitive performance. Consequently, we aimed to explore the association between high MCV level and cognitive function based on data from China Health and Retirement Longitudinal Study (CHARLS). We hypothesized that higher MCV levels than 100 fl might be associated with worse cognitive performance.

## Materials and methods

### Study population

The current study was based on the baseline data of CHARLS, which is conducted by the National School of Development at Peking University from 2011 to 2012. The current study

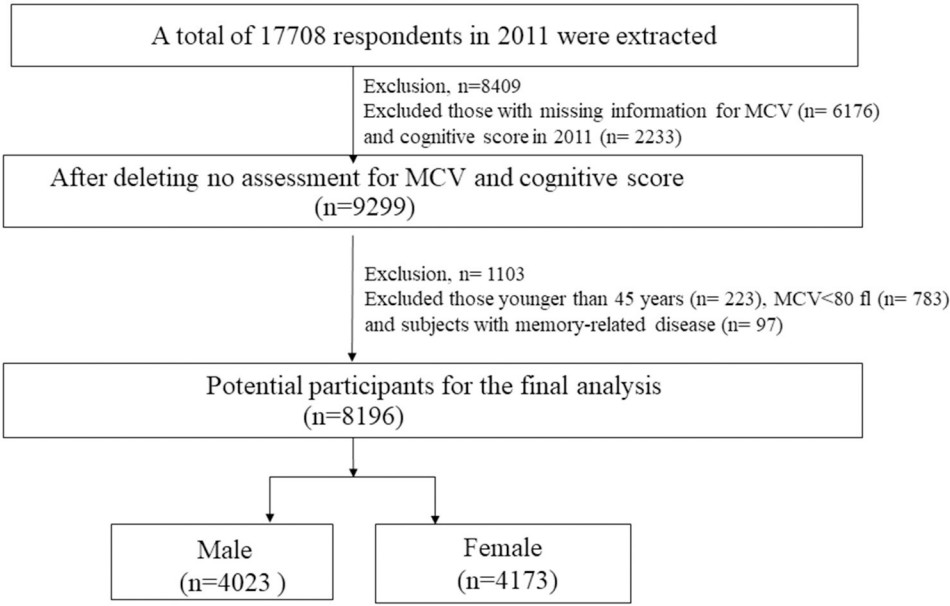

**Fig 1. Flowchart of recruited participants.**

was conducted according to the guidelines laid down in the Declaration of Helsinki, and all procedures involving study participants were approved by the biomedical ethics committee of Peking University. Written informed consent was obtained from all participants. The specified design of sampling methods for CHARLS has been described previously by Zhao et al. [23]. The flowchart of sample selection for this study was shown in Fig 1. A total of 17708 samples at baseline were extracted and 8409 participants were excluded for those lacking MCV (n = 6176) and cognitive score (n = 2233). Then a total of 1103 participants were excluded for those younger than 45 years old (n = 223), MCV<80 fl (n = 783), and subjects with memory-related disease (n = 97). Consequently, our final analysis compromised of 8,196 subjects. The subjects were further stratified into male (n = 4,023) and female (n = 4,173) group.

## Measurement of cognitive function

Two main assessments for assessing mental status and episodic memory were utilized to identify the cognitive function of participants by trained interviewers, which have been described in multiple previous studies [7, 24–27]. The mental status measurement was based on the Telephone Interview of Cognitive Status (TICS) and drawing a figure successfully. The TICS is one of the more practical ways of testing the respondents' mental status of cognition, which is seen as the telephone version of MMSE. The TICS contains 10 questions about today's date (year, month and day), the day of the week and season of the year, and to subtract 7 consecutive times (up to 5 times) from 100. A correct answer is equal to one point and the final score ranges from 0 to 10. Additionally, participants were asked to repaint a picture given by the interviewers and he/she could get 1 point if the participant has drawn the figure successfully. The second assessment was designed to assess episodic memory through immediate and delayed recall of 10 Chinese nouns. Ten unrelated Chinese words were read by interviewers only once, and each participant was required to recall of Chinese nouns immediately (immediate word recall) and to recall the same nouns again 4 minutes later (delayed word recall). The memory ability was evaluated by counting the average of how many words they recall correctly

in both immediate- and delayed-word recall tests, with the final score ranging from 0 to 10. In order to compare individual scores from different measurements, we used Z-scores to standardize different tests' scores based on the mean and standard deviation of the domain-specific cognitive score (i.e. scores for mental status and episodic memory, respectively) and global cognitive score (i.e. scores for the summation of mental status and episodic memory) from recruited subjects [28]. The lowest quartiles of Z score was derived for both global and domain specific cognitive score. We defined the respondents have poorer cognitive function if a Z-score < the lowest quartile (i.e. p25) of respective Z score.

## Assessment of MCV and covariates

For estimation of MCV, 8mL of venous fasting blood sample was collected and the detailed protocols for blood collection, storage and biomedical measurement have been described previously by Zhao et al. [23] and other studies [7, 24]. Each participant's MCV level was measured by medically-trained staff on automated analyzers available at the local hospitals or China Center for Disease Control and Prevention on automated analyzers. The calculation of MCV value is by multiplying the percent hematocrit by ten divided by the erythrocyte count [19]. We divided the subjects into normal (80≤MCV≤100 fl) and higher (MCV>100 fl) MCV groups.

Potential covariates including participants' socio-demographic factors (age, Hukou, and education level) and health-related variables [cigarette smoking and alcohol drinking status, body mass index (BMI), chronic diseases and blood biochemical index] were collected at the baseline survey. Hukou is the registration system in China created in 1955 to restrict internal population movement, especially rural-to-urban migration [29] and stratified Chinese residents into three groups: agriculture, non-agriculture and unified residency. Education level was classified as four mutually exclusive categories: illiterate, elementary school, middle school and high school or above. Cigarette smoking was categorized into current, former and never. Alcohol drinking status was categorized into excessive alcohol consumption or not. Excessive alcohol consumption was defined as ≥210 g alcohol/week for men and ≥140 g alcohol/week for women [30]. BMI was obtained directly from weight and height and calculated by the standard formula: kg/m$^2$. Chronic diseases including diabetes and hypertension were diagnosed by self-reported doctors' diagnosis and medication use. In addition to MCV, other blood biochemical measurements including hemoglobin and hematocrit were measured on the same device as MCV. The detection procedures of other blood bioassays were reported in multiple related studies [7, 24]. In brief, blood lipids consisting of low-density lipoprotein (LDL) cholesterol, high-density lipoprotein (HDL) cholesterol, total cholesterol (TC) and triglycerides (TG) were detected by enzymatic colorimetric test. Glycosylated hemoglobin (HbA1c) was measured by boronate affinity HPLC. Equation for estimated glomerular filtration rate (GRF) was calculated by the CKD-EPI creatinine equation [31]. Dyslipidemia was defined as TC≥240mg/dl or LDL-C≥160mg/dl or TG≥200mg/dl according to the "Guidelines for the Prevention and Treatment of Dyslipidemia in Adults in China (2016 Revised Edition)" [32].

## Statistical analysis

Continuous variables were expressed as Mean (SD) for normally distributed data and median (interquartile range, IQR) for non-normally distributed data, respectively. Categorical variables were expressed as number (percentage). Baseline characteristics were compared between the male and female subjects by performing Student's t-test and Wilcoxon test for normally and non-normally distributed continuous data, respectively. Additionally, Chi-square test was used for the comparison of categorical variables.

Multivariable linear regression models and logistic regression models were conducted to assess the relationship between MCV level and cognitive function. To assess potential confounding factors, covariate adjustments were used to gradually add demographic factors, health-related factors and blood biomedical factors into statistical models. The first statistical crude model (Model 1) was used to examine the raw association between MCV and cognitive function. The second statistical model (Model 2) was additionally included age, sex, Hukou and education level on the basis of Model 1. Model 3 was adjusted for health-related drinking status, smoking status, diabetes, hypertension, BMI and blood biochemical factors, including hemoglobin, dyslipidemia and GFR based on Model 2. To further investigate whether there was heterogeneity between male and females with varied MCV level, we have classified the subjects into 4 groups, i.e. female with 80≤MCV≤100 (low MCV), female with MCV>80 (high MCV), male with 80≤MCV≤100 (low MCV), and male with MCV>80 (high MCV). $I^2$ and Q statistic were used to examine the heterogeneity between studies via meta analysis. $I^2$<25% suggested little or no heterogeneity, 25–50% suggested moderate heterogeneity, and >50% was considered as high heterogeneity, respectively. As for Q statistic, p<0.1 indicated statistically significant [33]. Beta coefficient (β), standard error (SE) and *p* value for each variable in multiple linear regression models were presented. The analysis result of binary logistic regression is expressed as odds ratio (OR), 95% confidence interval (95% CI) and *p* value. All statistical processes were carried out using the Statistical Package for the Social Sciences (SPSS), version 22.0 (SPSS Inc, Chicago, IL, U.S.A.). Reported *p* values were two-tailed with a statistically significant level of $p < 0.05$.

## Results

### Baseline characteristics and cognitive function of recruited participants

Table 1 presented baseline characteristics and cognition measurement scores across male and female groups in 2011. A large part of the participants are rural households (80.2%). Only 11.5% of participants had high school or above education level. As for smoking status, men are more prone to consume cigarettes than women because more than 90% of female group members are never smoking. Female subjects have more excessive alcohol consumers than male subjects (p<0.005).The BMI of the female group was significantly higher than the male group (p<0.001). There were 489 diabetic patients and 2133 hypertensive patients, and the female group had significantly more participants afflicted with diabetes and hypertension than the male group (p<0.005). The hemoglobin level of women was significantly lower than men (p<0.001).

The median score of global cognitive function, episodic memory and mental status were significantly lower from female participants compared to male participants (p<0.001). The mean MCV of female respondents [mean (SD) = 91.29(5.54)] were significantly lower than the male subjects [mean (SD) = 93.50(6.23)] (p<0.001).

### Z-score of cognitive function

Table 2 demonstrated mean and standard deviation of cognitive Z scores for all participants stratified by MCV level and sex. Approximately 90% (n = 7440) of the examined subjects were in the normal MCV level (80≤MCV≤100 fl) and 10% (n = 756) were in the second status (MCV >100 fl). Moreover, more men (n = 521) were in the abnormal MCV level than women (n = 235). We defined Z-score of cognition (Cognition Z) = 0 as the average cognition score for total participants. The mean Z scores of global cognitive function and episodic memory for participants with MCV higher than 100 were significantly lower than those with normal MCV level for the whole subjects. When adjusted by sex, the mean Z scores of global cognitive

**Table 1. Basic characteristics of the CHARLS respondents in 2011 stratified by sex.**

| Variables | Total | Male | Female | p value |
|---|---|---|---|---|
| **Age (year)** (Mean ± SD) | 58.57 ± 9.05 | 59.21 ± 8.98 | 57.95 ± 9.07 | **<0.001**[b] |
| **Hukou,** n (%) | | | | **0.003**[a] |
| Agriculture | 6573 (80.2) | 3170 (78.8) | 3403 (81.6) | |
| Non-Agriculture | 1582 (19.3) | 832 (20.7) | 750 (18.0) | |
| Unified Residency | 38 (0.5) | 19 (0.5) | 19 (0.5) | |
| **Educational level,** n (%) | | | | **<0.001**[a] |
| Illiterate | 2051 (25.0) | 459 (11.4) | 1592 (38.2) | |
| Elementary school | 3428 (41.8) | 1874 (46.6) | 1554 (37.2) | |
| Middle school | 1771 (21.6) | 1086 (27.0) | 685 (16.4) | |
| High school and above | 946 (11.5) | 604 (15.0) | 342 (8.2) | |
| **Smoking status,** n (%) | | | | **<0.001**[a] |
| Current | 2611 (31.9) | 2360 (58.7) | 251 (6.0) | |
| Former | 793 (9.7) | 708 (17.6) | 85 (2.0) | |
| Never | 4790 (58.5) | 954 (23.7) | 3836 (91.9) | |
| **Drinking Status,** n (%) | | | | **0.004**[a] |
| Excessive alcohol consumption | 541 (6.6) | 233 (5.8) | 308 (7.4) | |
| **BMI (kg/m$^2$)** (Mean ± SD) | 23.67 ± 3.96 | 23.11 ± 3.65 | 24.2 ± 4.16 | **<0.001**[b] |
| **Diabetes,** n (%) | 489 (6.0) | 201 (5.0) | 288 (6.9) | **<0.001**[a] |
| **Hypertension,** n (%) | 2133 (26.0) | 989 (24.6) | 1144 (27.4) | **0.004**[a] |
| **Dyslipidemia,** n (%) | 1423 (17.4) | 694 (17.3) | 729 (17.5) | 0.413 [a] |
| **Blood Biochemical Index** (Mean ± SD) | | | | |
| Hemoglobin (g/dl) | 14.57 ± 2.11 | 15.31 ± 2.02 | 13.86 ± 1.96 | **<0.001**[b] |
| GFR (ml/min/1.73m$^2$) | 410.23 ± 239.20 | 411.56 ± 237.68 | 408.95 ± 240.69 | 0.690[b] |
| **Global cognitive function,** Median (IQR) | 11.5 (8.0 to 14.0) | 12.5 (9.5 to 14.5) | 10.5 (7.0 to 13.5) | **<0.001**[c] |
| **Episodic memory score,** Median (IQR) | 3.5 (2.5 to 4.5) | 3.5 (2.5 to 5.0) | 3.5 (2.5 to 4.5) | **<0.001**[c] |
| **Mental status score,** Median (IQR) | 8.0 (5.0 to 10.0) | 9.0 (6.0 to 11.0) | 7.0 (4.0 to 10.0) | **<0.001**[c] |
| **MCV (fl)** (Mean ± SD) | 92.37 ± 5.99 | 93.50 ± 6.23 | 91.29 ± 5.54 | **<0.001**[b] |

[a.] Chi-square test

[b.] Student's *t* test

[c.] Wilcoxon test

Hukou is the registration system in China created in 1955 to restrict internal population movement; BMI, body mass index; HbA1c, glycated hemoglobin; GRF, equation for estimated glomerular filtration rate; MCV, Mean Corpuscular Volume.

function, episodic memory and mental status were all significantly lower for those with MCV above 100 fl in comparison with subjects having normal MCV level for both male and female participants.

## Cross-sectional association between MCV and cognitive function

Multivariable linear regression model was firstly performed to explore the association between MCV level and global cognitive function, as well as domain-specific episodic memory and mental status. As shown in Table 3, for the whole subjects, MCV level was significantly negatively associated with global cognitive function (β = -0.038, -0.040 and -0.038 for model 1, 2 and 3, respectively, p<0.05) and episodic memory (β = -0.055, -0.035 and -0.036 for model 1, 2 and 3, respectively, p<0.05). Additionally, for male participants, higher MCV level was associated with reduced scores for both global cognitive function, episodic memory and mental status even adjusting for all additional covariates (model3) (β = -0.061, -0.051 and -0.049,

**Table 2. Mean and standard deviation of cognitive Z scores stratified by MCV levels.**

| Cognition Variable | Global cognitive function | | Episodic Memory | | Mental status | |
|---|---|---|---|---|---|---|
| | Mean ± SD | p value | Mean ± SD | p value | Mean ± SD | p value |
| **Total** | | | | | | |
| 80≤MCV≤100 (n = 7440) | 0.02 ± 1.67 | **<0.001** | 0.02 ± 1.00 | **<0.001** | 0.01 ± 1.00 | 0.159 |
| MCV >100 (n = 756) | -0.22 ± 1.61 | | -0.16 ± 0.99 | | -0.06 ± 0.98 | |
| **Male** | | | | | | |
| 80≤MCV≤100 (n = 3502) | 0.32 ± 1.53 | **<0.001** | 0.05 ± 0.96 | **<0.001** | 0.27 ± 0.89 | **0.001** |
| MCV >100 (n = 521) | -0.01 ± 1.52 | | -0.13 ± 0.98 | | 0.12 ± 0.91 | |
| **Female** | | | | | | |
| 80≤MCV≤100 (n = 3938) | -0.24 ± 1.75 | **<0.001** | -0.01 ± 1.03 | **0.001** | -0.23 ± 1.04 | **0.003** |
| MCV >100 (n = 235) | -0.68 ± 1.73 | | -0.22 ± 1.02 | | -0.46 ± 1.03 | |

Data are presented as mean ± SD. Z scores of domain-specific episodic memory and mental status, as well global cognitive function were derived based on the mean and standard deviation of respective cognitive scores.

respectively, p<0.05). After adjusted for all additional covariates, no association was observed between MCV level and mental status for the whole subjects, also there was no association between MCV level and global cognitive function, as well as episodic memory for female subjects.

Furthermore, binary logistic regression was conducted to confirm the association between MCV level and cognitive function (Table 4). After adjusting for potential confounding factors and being referenced to the normal MCV level, the higher MCV level (MCV>100 fl) was associated with poor global cognitive function (OR = 1.601; 95% CI = 1.198–2.139; p = 0.001), episodic memory (OR = 1.679; 95% CI = 1.281–2.201; p<0.001), and mental status (OR = 1.422; 95% CI = 1.032–1.959; p = 0.031) in the Model 3. When testing this association by sex after adjusting for all potential covariates, the significant relationship between higher MCV level with worse global cognitive function (OR = 1.885; 95% CI = 1.329–2.675; p<0.001), episodic memory (OR = 1.690; 95% CI = 1.211–2.358; p = 0.002) and mental status (OR = 1.544; 95% CI = 1.034–2.306; p = 0.034) was observed in the male subjects. In contrast, in female subjects,

**Table 3. Multiple linear regression model testing the association between MCV and cognitive function.**

| | Global cognitive function | | Episodic memory | | Mental status | |
|---|---|---|---|---|---|---|
| | β (95% CI) | p value | β (95% CI) | p value | β (95% CI) | p value |
| **Total (n = 8196)** | | | | | | |
| Model 1 | -0.038 (-0.020, -0.001) | **0.032** | -0.055 (-0.015, -0.003) | **0.002** | -0.009 (-0.007, 0.004) | 0.631 |
| Model 2 | -0.040 (-0.020, -0.003) | **0.010** | -0.035 (-0.011, 0.000) | **0.036** | -0.031 (-0.010, 0.000) | **0.047** |
| Model 3 | -0.038 (-0.019, -0.002) | **0.016** | -0.036 (-0.012, 0.000) | **0.036** | -0.027 (-0.010, 0.001) | 0.093 |
| **Male (n = 4023)** | | | | | | |
| Model 1 | -0.087 (-0.033, -0.009) | **0.001** | -0.071 (-0.018, -0.003) | **0.006** | -0.074 (-0.018, -0.003) | **0.004** |
| Model 2 | -0.062 (-0.026, -0.004) | **0.006** | -0.047 (-0.014, 0.000) | 0.051 | -0.056 (-0.015, -0.001) | **0.016** |
| Model 3 | -0.061 (-0.025, -0.004) | **0.008** | -0.051 (-0.015, -0.001) | **0.033** | -0.049 (-0.014, 0.000) | **0.037** |
| **Female (n = 4173)** | | | | | | |
| Model 1 | -0.053 (-0.033, -0.001) | **0.036** | -0.052 (-0.020, 0.000) | **0.040** | -0.036 (-0.016, 0.002) | 0.149 |
| Model 2 | -0.018 (-0.019, 0.007) | 0.390 | -0.023 (-0.013, 0.004) | 0.321 | -0.007 (-0.009, 0.007) | 0.742 |
| Model 3 | -0.016 (-0.019, 0.008) | 0.449 | -0.020 (-0.013, 0.005) | 0.402 | -0.007 (-0.010, 0.007) | 0.742 |

Model 1: Crude model. Model 2: adjusted for model 1+ age, sex, Hukou and education level. Model 3: adjusted for model 2+ drinking status, smoking status, diabetes, hypertension, hemoglobin, dyslipidemia, GFR and BMI.

**Table 4. Binary logistic regression model testing the association between MCV level and cognitive performance.**

|  | Global cognitive function | | Episodic memory | | Mental status | |
|---|---|---|---|---|---|---|
|  | OR (95% CI) | *p* value | OR (95% CI) | *p* value | OR (95% CI) | *p* value |
| **Total** | | | | | | |
| **Model 1** | | | | | | |
| 80≤MCV≤100 (*ref.*) | 1 (Reference) | | 1 (Reference) | | 1 (Reference) | |
| MCV>100 | 1.382 (1.072, 1.782) | **0.013** | 1.682 (1.306, 2.165) | **0.000** | 1.118 (0.850, 1.470) | 0.427 |
| **Model 2** | | | | | | |
| 80≤MCV≤100 (*ref.*) | 1 (Reference) | | 1 (Reference) | | 1 (Reference) | |
| MCV>100 | 1.621 (1.219, 2.157) | **0.001** | 1.674 (1.281, 2.188) | **0.000** | 1.444 (1.052, 1.981) | **0.023** |
| **Model 3** | | | | | | |
| 80≤MCV≤100 (*ref.*) | 1 (Reference) | | 1 (Reference) | | 1 (Reference) | |
| MCV>100 | 1.601 (1.198, 2.139) | **0.001** | 1.679 (1.281, 2.201) | **<0.001** | 1.422 (1.032, 1.959) | **0.031** |
| **Male** | | | | | | |
| **Model 1** | | | | | | |
| 80≤MCV≤100 (*ref.*) | 1 (Reference) | | 1 (Reference) | | 1 (Reference) | |
| MCV>100 | 1.884 (1.368, 2.596) | **<0.001** | 1.756 (1.283, 2.404) | **<0.001** | 1.558 (1.076, 2.256) | **0.019** |
| **Model 2** | | | | | | |
| 80≤MCV≤100 (*ref.*) | 1 (Reference) | | 1 (Reference) | | 1 (Reference) | |
| MCV>100 | 1.940 (1.376, 2.735) | **<0.001** | 1.715 (1.235, 2.381) | **0.001** | 1.647 (1.110, 2.445) | **0.013** |
| **Model 3** | | | | | | |
| 80≤MCV≤100 (*ref.*) | 1 (Reference) | | 1 (Reference) | | 1 (Reference) | |
| MCV>100 | 1.885 (1.329, 2.675) | **<0.001** | 1.690 (1.211, 2.358) | **0.002** | 1.544 (1.034, 2.306) | **0.034** |
| **Female** | | | | | | |
| **Model 1** | | | | | | |
| 80≤MCV≤100 (*ref.*) | 1 (Reference) | | 1 (Reference) | | 1 (Reference) | |
| MCV>100 | 1.273 (0.821, 1.975) | 0.281 | 1.790 (1.152, 2.780) | **0.010** | 1.295 (0.832, 2.017) | 0.252 |
| **Model 2** | | | | | | |
| 80≤MCV≤100 (*ref.*) | 1 (Reference) | | 1 (Reference) | | 1 (Reference) | |
| MCV>100 | 1.111 (0.674, 1.831) | 0.679 | 1.658 (1.043, 2.636) | **0.032** | 1.143 (0.684, 1.912) | 0.610 |
| **Model 3** | | | | | | |
| 80≤MCV≤100 (*ref.*) | 1 (Reference) | | 1 (Reference) | | 1 (Reference) | |
| MCV>100 | 1.112 (0.667, 1.855) | 0.685 | 1.729 (1.079, 2.770) | **0.023** | 1.160 (0.685, 1.964) | 0.581 |

Model 1: Crude model. Model 2: adjusted for model 1+ age, sex, Hukou and education level. Model 3: adjusted for model 2+ drinking status, smoking status, diabetes, hypertension, hemoglobin, dyslipidemia, GFR and BMI.

higher MCV level was only associated with poor episodic memory (OR = 1.729; 95% CI = 1.079–2.770; p = 0.023).

## Heterogeneity test via meta analysis

Our results suggested high heterogeneity existed for subjects with varied MCV level for both male and female subjects ($I^2$ = 64.8%, p = 0.059, $I^2$ = 60.4%, p = 0.080, and $I^2$ = 63.7%, p = 0.064 for global cognitive function, episodic memory and mental status, respectively), this further suggested that the association between MCV and cognitive function should be explored by sex.

## Discussion

In this large cross-sectional study on middle-aged and older Chinese residents, we found that higher MCV level (>100fl) was associated with poor global cognitive function, as well as

domain-specific episodic memory and metal status, but, surprisingly, after testing this correlation by sex, the MCV's main effect on these cognitive domains fully remained in the male group, in female subjects, only a significant association between higher MCV level and episodic memory was observed.

Existing studies demonstrated that erythrocyte volume seemed to be negatively associated with cognitive performance [20, 21]. For instance, Danon et al. [20] reported that that the elderly with smaller erythrocytes performed better in the delayed recall tasks. Additionally, a recent longitudinal study conducted by Gamaldo et al. [21] showed that higher baseline MCV levels were also significantly associated with accelerated rates of decline on tasks of global mental status and long delay memory. Similarly, based on data from UK Biobank dataset, there was a significant negative correlation for MCV with reaction time [12]. Our present findings demonstrated that higher MCV level (>100 fl) was associated with poor global cognitive function, and domain specific episodic memory and metal status, which is partially consistent with these previous studies. It is suggested that in Chinese subjects over 45 years old, higher MCV level might be associated poor cognitive performance.

There are several possible explanations for the association between higher MCV level and poorer cognitive function. Mohanty et al. [34] reported that 15% of erythrocytes in AD subjects are elongated and have altered membrane structure. Morphological changes of erythrocytes might result in decreased deformability, disordered physical state of membrane proteins, and oxidation imbalance [22, 35], consequently affecting cognitive function [22]. In addition, the relationship between larger MCV and the poor performance of cognition could be attributed to pathophysiological changes in carrying oxygen and glucose to cerebral tissues [36], thus adversely impacting on cognitive performance.

Currently, there was no evidence about the relationship between MCV level and cognitive function stratified by sex. When it comes to sex discrepancy for the association between anemia and cognitive function, a previous study conducted in USA found that there was a relationship between low or high hemoglobin with worse global cognition, which was greater in women compared to men [13]. In contrast, by using data from CHARLS, Qin et al. [7] recently demonstrated that there was a negative significant association between anemia and global cognitive function, episodic memory and TICS, which is independently from sex. Our study demonstrated that there might be sex-specific association between higher MCV level and global cognitive function, as well as domain-specific episodic memory and mental status. To be specific, the association between higher MCV level and poor global cognitive function, as well as domain-specific episodic memory and mental status might be more pronounced in male subjects than in female subjects. Generally, in non-demented subjects, women tend to outperform men in verbal-based episodic memory tasks [37], and a steeper age-associated episodic memory decline for males [38]. However, the female superiority in episodic memory would be decreasing with advancing age [39]. This might explain, at least partially the fact that the higher MCV level was associated with poor episodic memory was observed both in male and female subjects. The differences in sample size, participants' characteristics and cognitive measurements also might contribute to these sex-specific effects. The lack of the association between MCV level and global cognitive function, as well as mental status in female subjects might be explained by the hormone discrepancy and physiological status which leads to estrogen loss induced by menopause, as well as the cross-sectional study design. Nevertheless, our current findings suggest that for middle-aged and elderly Chinese male subjects with higher MCV level than 100 fl, special attention on cognitive function might be warranted to prevent cognitive decline or dementia. Meanwhile, it is also suggested that further studies are still required to clarify the sex-specific association between MCV level and cognitive performance both cross-sectionally and longitudinally.

There are some strengths and limitations deserving to be mentioned. Firstly, to the best of our knowledge, the current study is the very first study to explore the cross-sectional association between normal and higher MCV level with cognition in Chinese population. Secondly, our study has a relatively large sample size, which allows a much greater possibility of making reasonable conclusions and providing support for further investigations on the underlying association between MCV and cognitive performance. Besides, CHARLS offers a broad range of potential confounders including blood biomedical parameters measured by standardized protocols and rigid quality control [23, 40], social-demographic and health-related factors. However, this study also has some limitations. In the first place, the majority of the recruited participants are lowly educated and 82.1% of them hold rural Hukou, which might not be representative of the general Chinese population. For example, Jia et al. [3] reported that a greatly higher prevalence of dementia and AD was found in rural areas than in urban ones and they proposed an explanation for the urban-rural differences is education. Hence, the extrapolation of this study still requires caution. Second of all, a relatively limited number of cognitive domains were measured, the association of MCV level with other cognitive function tests' scores such as reaction time and prospective memory remains unclear. Thirdly, some other variables including vitamin $B_{12}$, folate level and thyroid function are also closely associated with MCV level [41], how these factors might affect the association between MCV and cognitive performance remain unanswered. Last but not least, we only explored the association between higher MCV level and cognitive performance cross-sectionally, whether high MCV level might play a causal role in adversely affecting cognitive function remain unclear.

## Conclusions

In conclusion, in this large population-based cross-sectional study, we demonstrated that the association between higher MCV level and poor episodic memory existed both in male and female subjects, while the association between higher MCV level and poor global cognitive performance, as well as mental status only existed in male subjects. Our study suggests that for middle-aged and elderly Chinese male subjects with MCV level higher than 100 fl, it is necessary for them to pay special attention to cognitive function, in order to prevent cognitive decline. Meanwhile, further studies are required to evaluate the association between MCV level and cognitive performance by considering sex into consideration both cross-sectionally and longitudinally.

## Acknowledgments

We thank the research team, field team and all the participants for the CHARLS.

## Author Contributions

**Data curation:** Yao Chen, Chen'Xi' Nan Ma, Zhan Gao.

**Formal analysis:** Lan Luo, Jieyun Yin.

**Writing – original draft:** Yao Chen, Chen'Xi' Nan Ma.

**Writing – review & editing:** Zengli Yu, Zhongxiao Wan.

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
