## [Decision Letter · Decision Letter 0]

22 Sep 2020

PONE-D-20-25060

The cross-sectional association between mean corpuscular volume level and cognitive function in Chinese over 45 years old: evidence from the China Health and Retirement Longitudinal Study

PLOS ONE

Dear Dr. Wan,

Thank you for submitting your manuscript to PLOS ONE. After careful consideration, we feel that it has merit but does not fully meet PLOS ONE’s publication criteria as it currently stands. Therefore, we invite you to submit a revised version of the manuscript that addresses the points raised during the review process.

We look forward to receiving your revised manuscript.

Kind regards,

Claudia K. Suemoto

Academic Editor

PLOS ONE

Journal Requirements:

Reviewers' comments:

Reviewer's Responses to Questions

**Comments to the Author**

1. Is the manuscript technically sound, and do the data support the conclusions?

Reviewer #1: Partly

Reviewer #2: Partly

2. Has the statistical analysis been performed appropriately and rigorously? 

Reviewer #1: Yes

Reviewer #2: N/A

3. Have the authors made all data underlying the findings in their manuscript fully available?

Reviewer #1: Yes

Reviewer #2: Yes

4. Is the manuscript presented in an intelligible fashion and written in standard English?

Reviewer #1: Yes

Reviewer #2: Yes

5. Review Comments to the Author

Reviewer #1: COMMENTS TO AUTHOR(S):

In this well-designed and well-written manuscript, Yao Chen, et al, examined the cross-sectional baseline association between mean corpuscular volume (MCV) and cognitive performance in the China Health and Retirement Longitudinal Study (CHARLS). A few points need clarification and improvement to assist in putting the study in context with current literature.

(1) INTRODUCTION

There are many more evidence suggesting that there was no association between anemia or low hemoglobin and cognitive impairment. You just mentioned 2 studies. I suggest that you also evaluate other studies in order to make it clear that the topic remains unfinished. For example: https://doi.org/10.1111/j.1440-1819.2012.02347.x, https://doi.org/10.1176/appi.neuropsych.19040088, https://doi.org/10.1017/S1041610211001724.

The choice of specifically selecting MCV values > 80 is not clear in the text. I also suggest adding a justification for the study's interest in focusing on high levels of MCV. Example: There is biological plausibility for this hypothesis due to the fact that larger red blood cells may have more difficulty passing through small capillaries, compromising to deliver adequate amounts of oxygen to cerebral tissues.

(2) METHODS

SAMPLE

It is not clear why a total of 2,120 participants were excluded due to missed formation on covariates of minor importance. The default is just to treat them as NA (missing values). So, I would like to recommend redoing the sample excluding only patients with no data on cognition (n=4,171), MCV (n=4,238), and those who didn`t attend the study`s criteria (younger than 45 (n= 223), subjects with memory-related disease (n= 88), MCV less than 80 (n=690)). Thus, I would suggest the use of a larger sample with 8,298 individuals and the management of the absence of data in the other variables such as NA.

VARIABLES

I notice the lack of important variables that are more related to cognitive performance: income, thyroid function, medications that could alter cognitive function (i.e., antipsychotic medications, antiparkinsonian agents, and anticonvulsants), major depressive disorder, cerebrovascular disease (i.e., history of ischemic attack, ateriosclerotic cerebrovascular disease, cerebral arteriosclerosis, and stroke), GFR rates (i.e., using CKD-EPI equation), vitamin B12, folate. If you have information about any of these, I strongly recommend using them. Otherwise, use what you have with some suggestions for improvement below.

I didn`t understand the value of the information about drinking status: current, former and never. In this form, I do not see any advantage to use drinking status in analysis. The most valuable information for cognition is if the individual has excessive alcohol use. It was defined as ≥210 g alcohol/week for men and ≥140 g alcohol/week for women.

It is not clear why a large number of biochemical variables with no specific advantage were chosen. Excessive or unnecessary information tends to weaken your analysis model. I suggest thinking about cleaning these polluting variables. I would highlight the importance of hemoglobin only, but it can also be chosen to be presented as an anemia Yes or No variable (according to WHO criteria). If you have data on thyroid function, folate and vitamin B12, I suggest including it. I highlight the importance of information on folate and vitamin b12 as it is a study that specifically analyzes high levels of MCV. Both vitamin B12 and folate cause macrocytosis and are nutrients that have been linked to neural health with evidence of affecting cognition by itself. That is, they are extremely relevant as potential confounders of your findings.

There is no need for glycemic or glycated hemoglobin data if you have information about diabetes in the population studied. There is no need for BPD or SBP if you have information about hypertension in the population studied. If you insist on presenting lipids in your analysis, I recommend condensing the information into a single variable: dyslipidemia (Yes or No). For renal function, I recommend to use GFR rates (i.e., using CKD-EPI equation). I do not recommend to give so much importance or use any of the others biochemical variable in analytic models.

ANALYSIS

I would like to suggest that you prefer to use the term "crude" instead of "raw" (line 184). I take this opportunity to recommend that Model 1 to be considered a Crude Model should not be adjusted for any variable.

Most important: Your analysis suffers from overadjustment and unnecessary adjustment. It can obscure a true effect or create an apparent effect when none exists.

Almost all respondents (99.3%) has got married. I would recommend to exclude the adjustment for marital status.

My personal recommendations:

- Model 1 (Crude Model): without any adjustment.

- Model 2: adjusted for age, sex, Hukou, education level. (+ income if you have this data)

- Model 3: Model 2 + adjusted for smoking status, BMI, diabetes, hypertension, hemoglobin (or anemia according to WHO criteria), dyslipidemia (according to lipids measures). (+ excessive alcohol use, + thyroid function, + GFR if you have this data).

Further models with your other variables can be presented in text as sensitivity analyzes.

RESULTS

Try to clear table 1 with only the relevant variables (as stated in the previous observations).

For table 3, it is better and more recommended to present the beta-coefficient results without SE, but with 95% CI.

If you don’t have data about important variables (i.e., income, thyroid function, vitamin B12, folate) it is also important to point out this is a limitation in your study.

Reviewer #2: In aging society, the research question is very relevant and worth of investigation. So my comments involves only the study design and the statical analysis.

1. The sample selection criteria reported in Figure 1 lead to the exclusion of all respondents lacking cognitive scores. This is a clear problem if those with severe cognitive decline are excluded from testing (as in HRS), leading to a downward bias (in absolute terms) in the association bettie MCV and cognition. The authors can test it by regressing a dummy which indicates whether the respondent lacks the cognitive scores on the MCV value.

2. The authors do not explain why they are using both OLS and logistic regression. I suspect that they want to test for the presence of non-linearity. Moreover, they do not explain well what outcome variable the use for the logistic regression. I assume from the end of page 6, it is a dummy for poor cognitive score set at the median value. If so, I strongly suggest to set a more reasonable threshold for poor cognition based on the neuropsychological literature. For instance, I would suggest the first quartile or quintile or a cognitive score one SD below the mean.

3. Regarding the MCV, I don't get why the authors excludes values below 80.

4. The selection of the control sounds arbitrary and also potentially endogenous, especially those included in Model 2 and 3. What is the logic behind? What are the determinant of the MCV?

4. As it stands, it not very clear why the results differ by sex. Have the author tried to test the difference using pool data and an interaction term between sex and MCV?

6. PLOS authors have the option to publish the peer review history of their article (what does this mean?). If published, this will include your full peer review and any attached files.

Reviewer #1: **Yes: **José Benedito Ramos Valladão Júnior

Reviewer #2: No

---

## [Author Response · Author response to Decision Letter 0]

16 Oct 2020

Dear Dr. Suemoto:

In regards to our Manuscript PONE-D-20-25060 entitled The cross-sectional association between mean corpuscular volume level and cognitive function in Chinese over 45 years old: evidence from the China Health and Retirement Longitudinal Study, we would like to thank the reviewers and the Editors for their thorough, insightful and thoughtful review of our manuscript. Below, we have addressed each of the comments in a point-by-point manner. Changes to the text of the manuscript are highlighted in RED in the revised version of the manuscript. 

Especially, we have corrected the analytic models per the 1st reviewer’s suggestion, we have also redefined the poor cognitive function as the Z score less than the lowest quartiles of respective Z score per the 2nd reviewer’s suggestion. All results have been updated and the overall trends of the findings remained. We have also made sure that our manuscript meets PLOS ONE's style requirements.

We feel the manuscript is much improved after revising based on the constructive comments from the reviewers and hope that you will find it acceptable for publication in Plos One. We are also willing to make any further revisions if required.

Regards,

Zhongxiao Wan, PhD, Professor

Department of Nutrition and Food Hygiene

School of Public Health, Soochow University

Reviewers’ and editors’ comments are listed below.

Reviewer #1: COMMENTS TO AUTHOR(S):

In this well-designed and well-written manuscript, Yao Chen, et al, examined the cross-sectional baseline association between mean corpuscular volume (MCV) and cognitive performance in the China Health and Retirement Longitudinal Study (CHARLS). A few points need clarification and improvement to assist in putting the study in context with current literature.

(1) INTRODUCTION

There are many more evidence suggesting that there was no association between anemia or low hemoglobin and cognitive impairment. You just mentioned 2 studies. I suggest that you also evaluate other studies in order to make it clear that the topic remains unfinished. For example: https://doi.org/10.1111/j.1440-1819.2012.02347.x, https://doi.org/10.1176/appi.neuropsych.19040088, https://doi.org/10.1017/S1041610211001724.

Response: Thank you very much for providing us more evidence in regards to the association of anemia or low hemoglobin with cognitive impairment. We have put these valuable references into our revised introduction section.

The choice of specifically selecting MCV values > 80 is not clear in the text. I also suggest adding a justification for the study's interest in focusing on high levels of MCV. Example: There is biological plausibility for this hypothesis due to the fact that larger red blood cells may have more difficulty passing through small capillaries, compromising to deliver adequate amounts of oxygen to cerebral tissues.

Response: We greatly appreciate this comment. Sentences have been added in the introduction section.

(2) METHODS

SAMPLE

It is not clear why a total of 2,120 participants were excluded due to missed formation on covariates of minor importance. The default is just to treat them as NA (missing values). So, I would like to recommend redoing the sample excluding only patients with no data on cognition (n=4,171), MCV (n=4,238), and those who didn`t attend the study`s criteria (younger than 45 (n= 223), subjects with memory-related disease (n= 88), MCV less than 80 (n=690)). Thus, I would suggest the use of a larger sample with 8,298 individuals and the management of the absence of data in the other variables such as NA.

Response: Thank you very much! We have followed your suggestion, and we have only excluded those with no data on MCV (6176), cognitive score (2233). We further exlcuded those who didn`t attend the study`s criteria (younger than 45 (n= 223), MCV less than 80 (n=783), and subjects with memory-related disease (n= 97). Eventually, a total of 4023 male subjects and 4173 famale subjects were included in the final analysis. Additionally, we have managed the absence of the other variables as missing values.

VARIABLES

I notice the lack of important variables that are more related to cognitive performance: income, thyroid function, medications that could alter cognitive function (i.e., antipsychotic medications, antiparkinsonian agents, and anticonvulsants), major depressive disorder, cerebrovascular disease (i.e., history of ischemic attack, ateriosclerotic cerebrovascular disease, cerebral arteriosclerosis, and stroke), GFR rates (i.e., using CKD-EPI equation), vitamin B12, folate. If you have information about any of these, I strongly recommend using them. Otherwise, use what you have with some suggestions for improvement below.

Response: Thank you very much! In regards to the important variables mentioned here, we have calculated GFR rates, however, we don’t have other factors including income, thyroid function, medications that could alter cognitive function (i.e., antipsychotic medications, antiparkinsonian agents, and anticonvulsants), major depressive disorder, cerebrovascular disease (i.e., history of ischemic attack, ateriosclerotic cerebrovascular disease, cerebral arteriosclerosis, and stroke), vitamin B12 and folate. We have added GFR in the new model for the improvement, we have also discussed the limitations for the absence of other factors in the revised discussion section.

I didn’t understand the value of the information about drinking status: current, former and never. In this form, I do not see any advantage to use drinking status in analysis. The most valuable information for cognition is if the individual has excessive alcohol use. It was defined as ≥210 g alcohol/week for men and ≥140 g alcohol/week for women.

Response: Thank you very much! We have reclassified the drinking status into excessive alcohol consumption or not based on your suggestion.

It is not clear why a large number of biochemical variables with no specific advantage were chosen. Excessive or unnecessary information tends to weaken your analysis model. I suggest thinking about cleaning these polluting variables. I would highlight the importance of hemoglobin only, but it can also be chosen to be presented as an anemia Yes or No variable (according to WHO criteria). If you have data on thyroid function, folate and vitamin B12, I suggest including it. I highlight the importance of information on folate and vitamin b12 as it is a study that specifically analyzes high levels of MCV. Both vitamin B12 and folate cause macrocytosis and are nutrients that have been linked to neural health with evidence of affecting cognition by itself. That is, they are extremely relevant as potential confounders of your findings.

Response: We greatly appreciate your comment. We totally agree that vitamin B12 and folate are closely associated with MCV level, however, our study don’t have these data. We have discussed this as one of our study limitations. Additionally, we have only included hemaglobin, GFR and dyslipidemia of the blood biochemical index in the adjusted models and excluded all other unrelated factors per your suggestion.

There is no need for glycemic or glycated hemoglobin data if you have information about diabetes in the population studied. There is no need for BPD or SBP if you have information about hypertension in the population studied. If you insist on presenting lipids in your analysis, I recommend condensing the information into a single variable: dyslipidemia (Yes or No). For renal function, I recommend to use GFR rates (i.e., using CKD-EPI equation). I do not recommend to give so much importance or use any of the others biochemical variable in analytic models.

Response: Thank you very much! In our analytic models, we have removed glycemic hemoglobin, DBP and SBP data. We have presented lipids data into dyslipidemia (yes or no) based on your suggestion. We have added GFR rates in our new model. We greatly appreciate your suggestion!

ANALYSIS

I would like to suggest that you prefer to use the term "crude" instead of "raw" (line 184). I take this opportunity to recommend that Model 1 to be considered a Crude Model should not be adjusted for any variable.

Response: Thank you very much! We have made Model 1 as a crude model.

Most important: Your analysis suffers from overadjustment and unnecessary adjustment. It can obscure a true effect or create an apparent effect when none exists.

Almost all respondents (99.3%) has got married. I would recommend to exclude the adjustment for marital status.

Response: Thank you very much! Marital status adjustment has been excluded.

My personal recommendations:

- Model 1 (Crude Model): without any adjustment.

- Model 2: adjusted for age, sex, Hukou, education level. (+ income if you have this data)

- Model 3: Model 2 + adjusted for smoking status, BMI, diabetes, hypertension, hemoglobin (or anemia according to WHO criteria), dyslipidemia (according to lipids measures). (+ excessive alcohol use, + thyroid function, + GFR if you have this data).

Further models with your other variables can be presented in text as sensitivity analyzes. 

Response: We greatly appreciate your comments. Followed with your suggestion, model 1 was the crude model. Model 2 was further adjusted for age, sex, Hukou and education level. We had no income information, thus income was not included in the model 2. Model 3 was further adjusted for smoking status, BMI, diabetes, hypertension, hemoglobin, dyslipidemia, excessive alcohol consumption or not, and GFR.

RESULTS

Try to clear table 1 with only the relevant variables (as stated in the previous observations).

Response: We greatly appreciate your comments. We have revised table 1 with only the relevant variables included.

For table 3, it is better and more recommended to present the beta-coefficient results without SE, but with 95% CI.

Response: We have revised table 3, 95% CI data has been provided.

If you don’t have data about important variables (i.e., income, thyroid function, vitamin B12, folate) it is also important to point out this is a limitation in your study.

Response: Thank you! We have discussed this as one of our study limitations.

Reviewer #2: In aging society, the research question is very relevant and worth of investigation. So my comments involves only the study design and the statical analysis.

1. The sample selection criteria reported in Figure 1 lead to the exclusion of all respondents lacking cognitive scores. This is a clear problem if those with severe cognitive decline are excluded from testing (as in HRS), leading to a downward bias (in absolute terms) in the association bettie MCV and cognition. The authors can test it by regressing a dummy which indicates whether the respondent lacks the cognitive scores on the MCV value.

Response: We totally agree with your concern. We have compared the MCV values for those with and without the cognitive score, as shown below. There was no difference for MCV levels between the two groups. Thus, we might not be worried about this issue.

The comparisons of MCV levels for those with and without cognitive score

 N Mean ± SD

Without Cognitive score 1898 92.92 ± 6.49

With Cognitive score 8196 92.37 ± 5.99

2. The authors do not explain why they are using both OLS and logistic regression. I suspect that they want to test for the presence of non-linearity. Moreover, they do not explain well what outcome variable the use for the logistic regression. I assume from the end of page 6, it is a dummy for poor cognitive score set at the median value. If so, I strongly suggest to set a more reasonable threshold for poor cognition based on the neuropsychological literature. For instance, I would suggest the first quartile or quintile or a cognitive score one SD below the mean. 

Response: Thank you very much! Generally, linear regression gives you a continuous output, but logistic regression provides a constant output. The combination of OLS and logistic regression will compensate with each other, and make the associations much clearer. It is very common for utilizing both OLS and logistic regression analysis to solve questions. In regards to the outcome variable for logistic regression analysis, we totally agree that setting a Z-score <0 might not be the best, we have corrected this. In our revised manuscript, we have set Z score < the lowest quartile (i.e. p25) as as poorer cognitive function per your suggestion.

3. Regarding the MCV, I don't get why the authors excludes values below 80.

Response: Thank you! In the revised introduction section, we have added background information why we only included those with MCV≥80 fl. In brief, larger red blood cells may have more difficulty passing through small capillaries, compromising to deliver adequate amounts of oxygen to cerebral tissues, there is biological plausibility that higher MCV level than the normal level might be asscociated with worse cognitive performance.

4. The selection of the control sounds arbitrary and also potentially endogenous, especially those included in Model 2 and 3. What is the logic behind? What are the determinant of the MCV?

Response: Thank you! As responded to the first reviewer, we have corrected our control variables with those only closely associated with MCV were included (i.e. Model 1: Crude model. Model 2: adjusted for model 1+ age, sex, Hukou and education level. Model 3: adjusted for model 2+ drinking status, smoking status, diabetes, hypertension, hemoglobin, dyslipidemia, GFR and BMI.). It should be realized that some other variables which are also closely related to MCV including vitamin B12, folate and income were not controlled in our analysis because we couldn’t get these data. We have discussed this as one of our study limitation.

5. As it stands, it not very clear why the results differ by sex. Have the author tried to test the difference using pool data and an interaction term between sex and MCV?

Response: We greatly appreciate this comment. We have utilized the multivariate logistic regression model to test the interaction term between sex and MCV. As shown in newly added table 5, there was no interaction between MCV and sex in fully adjusted model, it is suggested that both sex and MCV affected cognitive function independently.

---

## [Decision Letter · Decision Letter 1]

9 Nov 2020

PONE-D-20-25060R1

The cross-sectional association between mean corpuscular volume level and cognitive function in Chinese over 45 years old: evidence from the China Health and Retirement Longitudinal Study

PLOS ONE

Dear Dr. Wan,

Thank you for submitting your manuscript to PLOS ONE. After careful consideration, we feel that it has merit but does not fully meet PLOS ONE’s publication criteria as it currently stands. Therefore, we invite you to submit a revised version of the manuscript that addresses the points raised during the review process.

We look forward to receiving your revised manuscript.

Kind regards,

Claudia K. Suemoto

Academic Editor

PLOS ONE

Additional Editor Comments (if provided):

Please, address the concern from Reviewer 2.

Reviewers' comments:

Reviewer's Responses to Questions

**Comments to the Author**

1. If the authors have adequately addressed your comments raised in a previous round of review and you feel that this manuscript is now acceptable for publication, you may indicate that here to bypass the “Comments to the Author” section, enter your conflict of interest statement in the “Confidential to Editor” section, and submit your "Accept" recommendation.

Reviewer #1: All comments have been addressed

Reviewer #2: (No Response)

2. Is the manuscript technically sound, and do the data support the conclusions?

Reviewer #1: Yes

Reviewer #2: Yes

3. Has the statistical analysis been performed appropriately and rigorously? 

Reviewer #1: Yes

Reviewer #2: Yes

4. Have the authors made all data underlying the findings in their manuscript fully available?

Reviewer #1: Yes

Reviewer #2: Yes

5. Is the manuscript presented in an intelligible fashion and written in standard English?

Reviewer #1: Yes

Reviewer #2: Yes

6. Review Comments to the Author

Reviewer #1: In my opinion, the revision carried out by Wan and collaborators met the need for clarification, adjustment and improvement of methodology and analysis.

Stylistically, I personally prefer to just emphasize the central results and main tables of the study. Thus, optionally, all the section "Combined effect of MCV and sex on the prediction of cognitive performance" and table 5 can be eliminated. Leaving only described in the text this piece: “As a sensitivity analysis, we have utilized the multivariate logistic regression model to test the interaction term between sex and MCV. No combined effect was observed between MCV and sex. It is suggested that both sex and MCV affected cognitive function independently”.

My compliments to Wan and collaborators to considering our comments, doing a great revision work and an excellent final article.

Reviewer #2: I'm fine with review which almost my comments.

The only final concern is how to reconcile the heterogeneity by sex in T4 with the evidence in T5 which suggest no heterogeneity, unless I missed something. In other words, the fully interacted model (T4) gives different results from the model with the sex interaction term (T5).

7. PLOS authors have the option to publish the peer review history of their article (what does this mean?). If published, this will include your full peer review and any attached files.

Reviewer #1: No

Reviewer #2: No

---

## [Author Response · Author response to Decision Letter 1]

15 Nov 2020

Dear Dr. Suemoto:

In regards to our Manuscript PONE-D-20-25060R1 entitled The cross-sectional association between mean corpuscular volume level and cognitive function in Chinese over 45 years old: evidence from the China Health and Retirement Longitudinal Study, we would like to thank the reviewers and the Editors for their thorough, insightful and thoughtful review of our manuscript. Below, we have addressed each of the comments in a point-by-point manner. Changes to the text of the manuscript are highlighted in BLUE in the revised version of the manuscript. 

Specially, both of the reviewers were concerned about table 5. We have used meta analysis to explore the potential heterogeneity (see below response to #2 reviewer in detail, our results suggested high heterogeneity existed for subjects with varied MCV level for both male and female subjects, this further supported that the association between MCV and cognitive function should be explored by sex. We think it might not be necessary to include this section of results in the final manuscript, and we have also removed table 5 per the first reviewer’s suggestion.

We feel the manuscript is much improved after revising based on the constructive comments from the reviewers and hope that you will find it acceptable for publication in Plos One.

Regards,

Zhongxiao Wan, PhD, Professor

Department of Nutrition and Food Hygiene

School of Public Health, Soochow University

Reviewers’ and editors’ comments are listed below, followed by our responses in bold-faced type.

Reviewer #1: In my opinion, the revision carried out by Wan and collaborators met the need for clarification, adjustment and improvement of methodology and analysis.

Stylistically, I personally prefer to just emphasize the central results and main tables of the study. Thus, optionally, all the section "Combined effect of MCV and sex on the prediction of cognitive performance" and table 5 can be eliminated. Leaving only described in the text this piece: “As a sensitivity analysis, we have utilized the multivariate logistic regression model to test the interaction term between sex and MCV. No combined effect was observed between MCV and sex. It is suggested that both sex and MCV affected cognitive function independently”.

My compliments to Wan and collaborators to considering our comments, doing a great revision work and an excellent final article.

Response: Thank you very much! We have followed your suggestion, and removed table 5, with the results being described in the text. It should be mentioned that we have used meta analysis to explore the potential heterogeneity (see below response to #2 reviewer in detail), with heterogeneity were observed. This further confirmed that we should explore the association between MCV and cognitive function by sex.

Reviewer #2: I'm fine with review which almost my comments.

The only final concern is how to reconcile the heterogeneity by sex in T4 with the evidence in T5 which suggest no heterogeneity, unless I missed something. In other words, the fully interacted model (T4) gives different results from the model with the sex interaction term (T5).

Response: Thank you very much! We totally agree with your confusion, and we apologize for not solving this issue clearly in the first revision. In this revised manuscript, we have classified the subjects into 4 groups, i.e. female with 80≤MCV≤100 (low MCV), female with MCV>80 (high MCV), male with 80≤MCV≤100 (low MCV), and male with MCV>80 (high MCV). We have used meta analysis to explore the potential heterogeneity, I2<25% suggested little or no heterogeneity, 25–50% suggested moderate heterogeneity, and >50% was considered as high heterogeneity, respectively. As for Q statistic, p<0.1 indicated statistically signiﬁcant. As shown in below figure 1, our results suggested high heterogeneity existed for subjects with varied MCV level for both male and female subjects, this further suggested it is necessary to look at the association between MCV and cognitive function by sex. We have mentioned this part of results in the text and we have also removed table 5 per the first reviewer’s suggestion.

---

## [Editor Report · Decision Letter 2]

18 Nov 2020

The cross-sectional association between mean corpuscular volume level and cognitive function in Chinese over 45 years old: evidence from the China Health and Retirement Longitudinal Study

PONE-D-20-25060R2

Dear Dr. Wan,

We’re pleased to inform you that your manuscript has been judged scientifically suitable for publication and will be formally accepted for publication once it meets all outstanding technical requirements.

Kind regards,

Claudia K. Suemoto

Academic Editor

PLOS ONE
---

## [Editor Report · Acceptance letter]

23 Nov 2020

PONE-D-20-25060R2 

The cross-sectional association between mean corpuscular volume level and cognitive function in Chinese over 45 years old: evidence from the China Health and Retirement Longitudinal Study 

Dear Dr. Wan:

I'm pleased to inform you that your manuscript has been deemed suitable for publication in PLOS ONE. Congratulations! Your manuscript is now with our production department. 

Kind regards, 

on behalf of

Dr. Claudia K. Suemoto 

Academic Editor

PLOS ONE